# SNAREs define targeting specificity of trafficking vesicles by combinatorial interaction with tethering factors

Seiichi Koike[1] & Reinhard Jahn[1]

Membrane traffic operates by vesicles that bud from precursor organelles and are transported to their target compartment where they dock and fuse. Targeting requires tethering factors recruited by small GTPases and phosphoinositides whereas fusion is carried out by SNARE proteins. Here we report that vesicles containing the Q-SNAREs syntaxin 13 (Stx13) and syntaxin 6 (Stx6) together are targeted to a different endosomal compartment than vesicles containing only Stx6 using injection of artificial vesicles. Targeting by Stx6 requires Vps51, a component of the GARP/EARP tethering complexes. In contrast, targeting by both Stx6 and Stx13 is governed by Vps13B identified here as tethering factor functioning in transport from early endosomes to recycling endosomes. Vps13B specifically binds to Stx13/Stx6 as well as to Rab14, Rab6, and PtdIns(3)P. We conclude that SNAREs use a combinatorial code for recruiting tethering factors, revealing a key function in targeting that is independent of SNARE pairing during fusion.

[1] Department of Neurobiology, Max Planck Institute for Biophysical Chemistry, Göttingen 37077, Germany. Correspondence and requests for materials should be addressed to R.J. (email: rjahn@gwdg.de)

In eukaryotic cells, communication between organelles is mostly mediated by trafficking vesicles that bud from precursor compartments and are transported along cytoskeletal tracks toward the destination membrane where they dock and fuse[1]. To ensure that each trafficking vesicle is targeted to the correct compartment, unique identifiers (molecular "zip-codes") are incorporated into vesicles as they bud off a donor organelle. Three types of macromolecules have emerged as major zip code candidates. They include (i) small GTPases mostly of the Rab and Arf/Arl families[2,3], (ii) phosphorylated variants of the membrane lipid phosphatidylinositol (PtdIns phosphates)[4], and (iii) soluble N-ethylmaleimide-sensitive factor attachment protein receptor (SNARE) proteins that execute membrane fusion in the secretory pathway[5]. Together, they orchestrate a multilayered recognition system at the target site to ensure targeting specificity.

The first layer is represented by vesicle tethering that captures the trafficking vesicle and brings it close to the target membrane. It is mediated by tethering factors that are transiently recruited from the surrounding cytoplasm and released when the trafficking step is completed[6–8]. Tethering factors are not only necessary for targeting but also appear to be sufficient as shown in elegant relocation assays where tethering factors were artificially mislocalized to mitochondria, resulting in mistargeting of the corresponding vesicles[9,10]. So far, ~20 different tethering factors are known that fall into two major groups: multi-subunit tethering complexes (MTCs) and homodimeric coiled-coil proteins. Both class of tethering factors share certain characteristics including specific binding to zip code molecules (see below) or coat proteins, the ability to bridge membranes, and to regulate membrane[6,7].

The second layer of the recognition system is thought to be represented by the specificity of SNARE assembly between the tethered vesicle and the target membrane, which drives membrane fusion. Mammalian cells contain between 35 and 40 different SNAREs, and SNARE complexes involved in different intracellular fusion steps differ in their composition[5]. It is still debated to which extent the combinatorial composition of the respective SNARE complex encodes specificity since some SNAREs can functionally substitute for each other as long as they belong to the same subfamily[11,12]. However, there is general agreement that pairing specificity of SNAREs, despite some promiscuity, provides a final checkpoint for the accurate delivery of trafficking vesicles[13,14].

Thus, there appears to be a division of tasks between the first and second layer of recognition among the three classes of zip code molecules. Vesicle capture by tethering factors is thought to be mainly orchestrated by the small GTPases that can be switched on and off. In the GTP-bound "on" form, transport vesicles are specifically recognized by tethering factors. Moreover, in many cases, binding is enhanced by phosphorylated variants of the membrane lipid phosphatidylinositol (PtdIns phosphates)[4]. The seven known phosphoinositides as well as Rab GTPases are selectively associated with defined endomembrane compartments, thus providing identity signals that are recognized by tethering factors and other effectors[15]. Simultaneous binding to Rabs and PtdIns phosphates thus stabilizes membrane association via tethering factors[16,17]. Note that in some cases, the recognition may be indirect and involve additional proteins. For instance, the Dsl1 complex and GCC185 tethering factors recognize COPI and the AP-1 complex, respectively[18,19], which both are recruited via active Arf GTPases[3]. Thus, small GTPases and PtdIns phosphates are considered as prime players in defining targeting specificity via recruitment of tethering factors, whereas the role of SNAREs is hitherto thought to be confined to the final fusion step.

Intriguingly, certain tethering factors are also known to bind to SNARE proteins, and only subsets of SNAREs appear to be involved. As far as known, binding is mainly mediated by the N-terminal domains of SNAREs that are localized upstream of the SNARE motifs required for fusion[20]. Some of these interactions are rather stable, allowing for crystallization of the complex such as the yeast Dsl1-complex[21] or the complex between the N-terminal domain of syntaxin 6 (Stx6) and Vps51, a component of the multi-subunit GARP (golgi-associated retrograde protein)/EARP (endosome-associated recycling protein) tethering complexes[22]. While such interactions may further stabilize the binding of tethering factors to trafficking vesicles, SNARE binding is primarily thought to promote the assembly of specific and functional SNARE complexes after tethering[20]. SNAREs are generally not considered to play a role in targeting due to their broad distribution among intracellular membranes. As integral membrane proteins they can only be returned by membrane traffic to their site of action and are present on all vesicles of the respective recycling pathway[5]. Thus, they lack the stage-specificity of the GTPases and PtdIns phosphates, which is needed for distinguishing each individual trafficking step, but a rigorous test of the involvement of SNAREs has so far not been feasible.

Recently, we have made the surprising observation that injection of artificial vesicles results in targeting and fusion of the liposomes with endogenous early endosomes if they contain the SNAREs mediating fusion of early endosomes (syntaxin 13 (Stx13), Stx6, vti1a, and VAMP4) as the only proteins[23]. These observations show that an otherwise "naïve" vesicle devoid of PtdIns phosphates and GTPases only requires SNARE proteins to be phased into the correct domain of the secretory pathway. In the present study, we have used the same approach to shed light on the underlying molecular mechanisms. Using the early endosomal SNAREs as example, we show that targeting specificity is mediated by cooperative capturing of SNARE-binding tethering factors in a combinatorial manner. These include the Vps51, a component of the GARP complex that is recruited by Stx6, and Vps13B, a large monomeric protein with unclear function that is recruited jointly by Stx6 and Stx13 13. We show here that Vps13B functions as a tethering factor in vesicle traffic between early and recycling endosomes. Our data suggest that at least some SNAREs can effectively recruit tethering proteins that alone or in combination define intracellular targeting.

## Results

**Liposomes containing endosomal SNAREs are correctly targeted.** To assess targeting of vesicles injected into HeLa cells, we determined co-localization of the fluorescence-labeled vesicles with a set of marker proteins 5 min after injection (see also ref. [23]). As depicted in Fig. 1a, these markers include APPL1 (adaptor protein, phosphotyrosine interaction, PH domain, and leucine zipper containing 1) for endocytic vesicles, EEA1 (early endosome antigen 1) for early endosomes, transferrin (Tfn) for early/recycling endosomes, M6PR (cation-independent mannose-6-phosphate receptor) for golgi/endosome transport vesicles, LBPA (lysobisphosphatidic acid) for late endosomes, LAMP1 (lysosomal-associated membrane protein 1) for lysosomes, GM130 for cis-Golgi, Golgin97 for the trans-Golgi network (TGN), and PDI (protein disulfide isomerase) for the endoplasmic reticulum. To quantify the degree of co-localization, the center of all spots in both channels was calculated and the distance of every injected vesicle to the closest particle was measured[23]. As a control for accidental overlap, we carried out experiments in which cells were kept at 4 °C.

As reference for our liposome experiments, we first injected enriched fractions of labeled early/recycling and late endosomes. To this end, HeLa cells were either preincubated with Alexa488-Tfn to label early/recycling endosomes, and transfected with

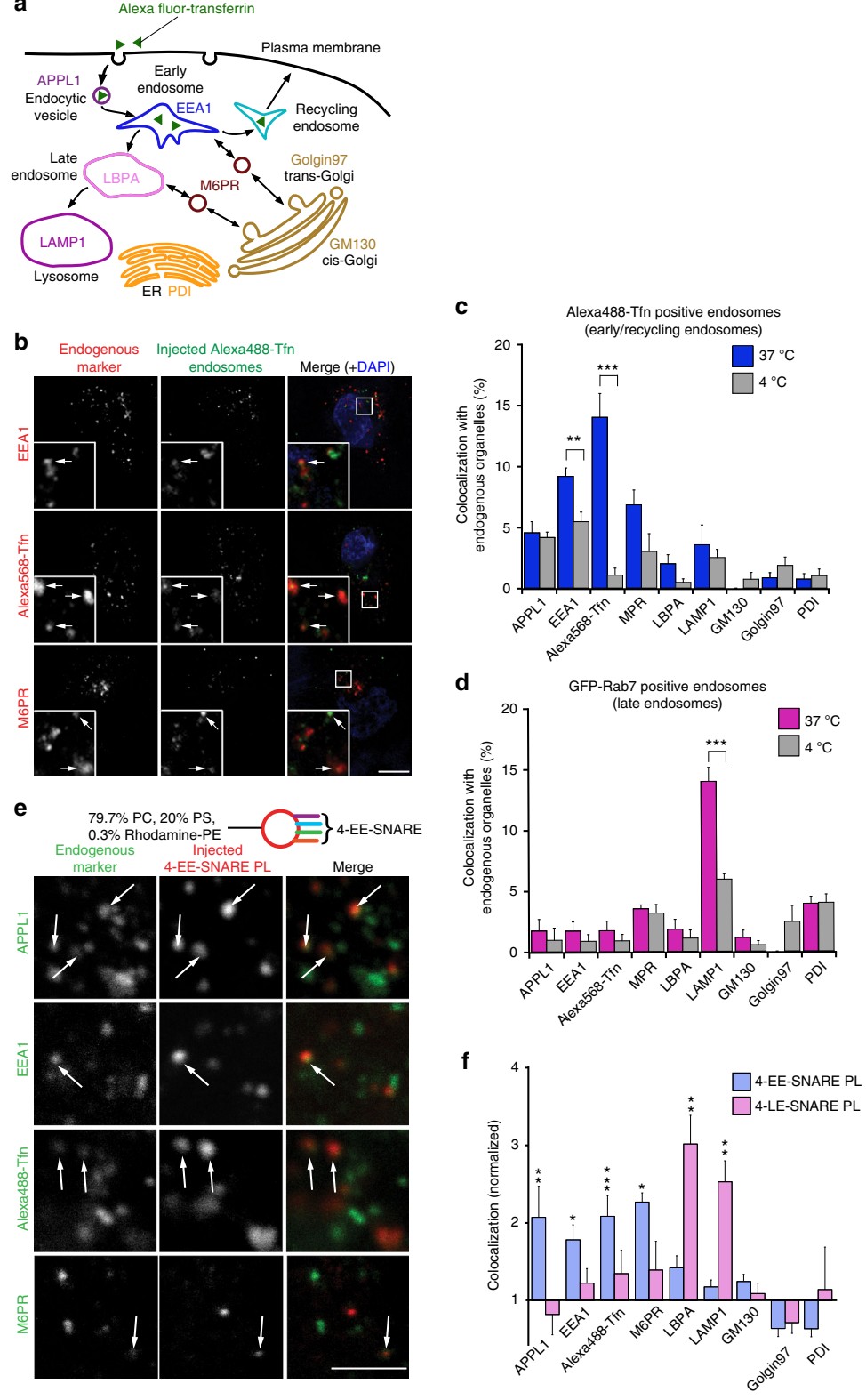

GFP-Rab7 to label late endosomes. Endosomes were then enriched using standard density gradient fractionation of post-nuclear supernatants (Supplementary Fig. 1a–d). Although the injected early/recycling endosomes are expected to contain at least some of these markers, the overlap between injected and endogenous EEA1/Tfn-positive endosomes, but not with other organelles, was much higher than in the 4 °C condition (Fig. 1b, c and Supplementary Fig. 2a), confirming our previous

observation[23]. Note that the injected early/recycling endosomes were functionally active: when endosomes derived from cells expressing a Tfn receptor (TfnR) containing a luminal GFP-tag were injected, surface exposure of the Green fluorescent protein (GFP) -tag was observable few minutes after injection (Supplementary Fig. 1e–h). Similarly, injection of late endosomes containing GFP-Rab7 resulted in specific and significant co-localization with LAMP1 that was reduced at 4 °C (Fig. 1d and

**Fig. 1** Liposomes containing endosomal SNAREs are targeted to their endogenous counterparts. **a** Schematic overview showing the organelle-specific markers used in this study. See text for abbreviations. **b** Representative confocal microscopy images of HeLa cells fixed 5 min after injection of Alexa Fluor 488-Tfn-labeled endosomes and DAPI (injection marker). Cells were immunolabeled for organelle markers except Alexa Fluor 568-Tfn that was internalized into the cells during microinjection for 5 min. Inserts show high magnifications of the boxed areas. White arrows indicate co-localization. Scale bar, 10 μm. **c** Co-localization between injected early endosomes and endogenous markers (in % of all labeled injected endosomes). In this and all subsequent figures, the data show mean values ± S.E.M. of 3–15 independent experiments. At least 100 injected vesicles were analyzed for the colocalization with each organelle marker in an experiment of microinjection. Scale bar, 10 μm. Stars indicating significance: $*P < 0.05$, $**P < 0.01$, $***P < 0.001$, all determined by unpaired $t$-test. **d** Co-localization between injected late endosomes and endogenous markers (in % of injected endosomes). **e** Representative confocal images of cells fixed 5 min after injection of proteoliposomes (PLs) reconstituted with early endosomal SNAREs (4-EE-SNARE PL). White arrows indicate co-localization between injected liposomes and organelle markers. Scale bar, 2.5 μm. **f** Co-localization between injected liposomes containing either four early endosomal SNAREs (4-EE-SNARE PL, light blue), or four late endosomal SNAREs (4-LE-SNARE PL) (light pink), respectively, and endogenous markers. Values were normalized to the degree of co-localization observed in control injections using protein-free liposomes

Supplementary Fig. 2b), showing that specific targeting occurs. Thus, isolated endosomes retain their targeting information during isolation.

Next, we injected artificial vesicles with a simple lipid composition (79.7% phosphatidylcholine, 20% phosphatidylserine, and 0.3% Rhodamine-PE) containing either the four SNAREs mediating early endosome fusion including Stx13, vti1a, Stx6, and VAMP4 (referred to as EE-SNAREs)[24–26] or the four SNAREs mediating late endosome fusion including syntaxin 7, vti1b, syntaxin 8, and VAMP8 (referred to as LE-SNAREs)[27] (note that each of these SNAREs contains a single C-terminal transmembrane domain). Injection of EE-SNARE liposomes resulted in a significantly higher co-localization with endocytic vesicles, early/recycling endosomes, and M6PR-containing vesicles than that of protein-free liposomes injected as control (Fig. 1e and Supplementary Fig. 2c–e). For better comparison, quantification data were normalized to the degree of co-localization observed in control injections using protein-free liposomes (Fig. 1f). The pattern (light blue bars in Fig. 1f) was similar to that observed after injection of Tfn-labeled endosomes (blue bars in Fig. 1c) except a somewhat broader targeting specificity of the EE-SNARE liposomes. In contrast, LE-SNARE liposomes showed a strong preference for late endosomes and lysosomes (light pink bars in Fig. 1f and Supplementary Fig. 2f) that was comparable to that of injected late endosome fractions (magenta bars in Fig. 1c).

**Targeting specificity is governed differentially by Stx13 and Stx6.** In the following experiments, we focused on the early endosomal SNAREs in order to dissect which of the SNAREs are responsible for the targeting of the artificial vesicles. As shown in Fig. 2a, omission of the R-SNARE VAMP4 from the liposomes had no effect on the targeting specificity. Next, we analyzed liposomes containing two of the three Q-SNAREs (all three combinations, Fig. 2b), and finally only individual SNAREs (Fig. 2c). Here, an interesting dissociation was observable: vesicles containing Stx6 and Stx13 again behaved similar to liposomes containing the complete set except lack of co-localization with APPL1. In contrast, liposomes containing only Stx6 co-localized rather selectively with M6PR-containing vesicles rather than with early endosomal markers, regardless of whether it was alone or combined with vti1a. The only other combination showing a similar preference but to a lesser extent (below significance) was Stx13 together with vti1a. In all other cases, no significant co-localization was observed.

Two conclusions can be drawn from these findings. First, only two of the tested four SNAREs have a significant role in vesicle targeting although there are hints of an influence of vti1a (APPL-co-localization). Second, Stx6 is the only SNARE that mediates targeting in the absence of any partner SNAREs but with a targeting specificity that is different from that seen when Stx13 is present. Thus, Stx13 influences targeting only in combination with Stx6, changing the targeting specificity mediated by Stx6

alone, suggesting that vesicles containing both Stx13 and Stx6 (Stx6+/Stx13+) follow different routes than vesicles containing only Stx6 (Stx6+/Stx13−). Using transfection of Hela cells with mRFP-Stx6 and GFP-Stx13, respectively, we therefore analyzed the intracellular distribution of these proteins in detail and determined their relationship to early endosomes involved in Tfn recycling and to endosomes carrying M6PR. In this analysis, we focused on the peripheral regions of the transfected cells where individual vesicles can be better resolved than in the densely labeled perinuclear region. Quantification revealed that practically all (95.3%) Stx13-positive vesicles ($n = 212$, from three independent experiments) were positive for Stx6 (Stx6+/Stx13+). Conversely, a majority (74.0 %) but not all Stx6-positive vesicles ($n = 273$) were positive for Stx13, i.e. the cells contained a significant proportion of vesicles positive for Stx6 but negative for Stx13 (Stx6+/Stx13−) (Fig. 3a). Next we examined co-localization of the SNAREs with Tfn and M6PR. Stx6+/Stx13+ vesicles showed major overlap with Tfn (93.5%). Overlap with M6PR was lower but still seen with the majority of this population (56.0%). In contrast, only an almost negligible proportion (2.8%) of the Stx6+/Stx13− vesicles were Tfn positive, whereas 14.8% of them were positive for M6RP.

To shed more light on the functional differentiation between Stx6+/Stx13+ and Stx6+/Stx13− vesicles, we asked whether these two SNAREs are differentially sorted at the level of the endosomal recycling compartment. Such sorting occurs at dynamic tubular extensions as shown, for instance by the localization of the TfnR (a cargo of recycling pathway) and cholera toxin B-subunit (CTxB) (a cargo for the TGN) to different tubules. Both tubules emanate from Tfn and CTxB-positive globular precursors of the endosomal recycling compartment, with the transport vesicles budding off at the tip of tubular extensions[28].

To this end, we allowed cells expressing mRFP-Stx6 and GFP-Stx13 to internalize either labeled Tfn or labeled CTxB and then monitored segregation of SNAREs and cargoes with time-lapse imaging (Fig. 3b–d) (note that for CTxB, Vero cells were used due to weak internalization in HeLa cells). Tubules positive for Alexa Fluor 633-Tfn were also positive for GFP-Stx13 and mRFP-Stx6 (Fig. 3b). However, tubular domains recruiting CTxB were positive for Stx6, but negative for Stx13, in contrast to the base of the tubule that contained Stx13 (Fig. 3c).

To confirm that Stx13 is sorted away from CTxB-containing vesicles derived from recycling endosomes, we internalized Alexa Fluor 647-CTxB and Alexa Fluor 568-Tfn in GFP-Stx13 transfected cells and monitored vesicle budding using time-lapse imaging. Figure 3d shows an example of a CTxB positive-transport vesicle that originates from a Tfn-, CTxB-, and Stx13-positive precursor but excludes Tfn and Stx13. We conclude that both Stx13 and Stx6 operate jointly in the recycling pathway of Tfn-positive endosomes, whereas vesicles containing Stx6 but devoid of Stx13 operate in the M6RP- and CtxB-positive retrograde pathway (Fig. 3e).

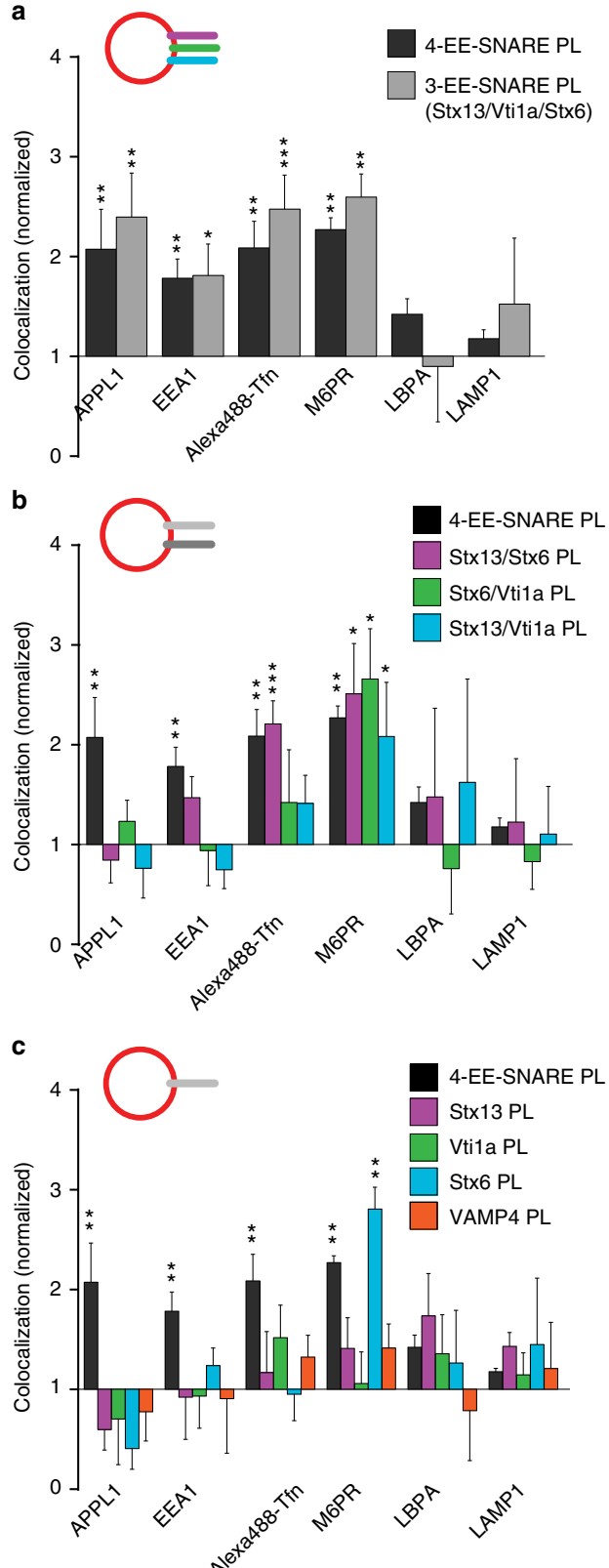

**Fig. 2** Targeting specificity depends on specific SNARE combinations. **a** Liposomes containing only the three early endosomal Q-SNAREs Stx13, Stx6, and vti1a (3-EE-SNARE PL) show the same targeting specificity as liposomes containing the complete set (4-EE-SNARE PL). All values were normalized to the degree of co-localization observed in control injections using protein-free liposomes. **b**, **c** Targeting specificity of injected liposomes containing either various combinations of two early endosomal SNAREs (**b**) or individual SNAREs (**c**). For comparison, the results obtained for liposomes containing the complete set of early endosomes (4-EE-SNARE PL) are shown. Note that no specific targeting was observed to vesicles containing Golgin97 or PDI (not shown). $*P < 0.05$, $**P < 0.01$, $***P < 0.001$, all determined by unpaired $t$-test

liposomes containing Stx6 with cytosol, separated the liposomes by flotation gradient centrifugation, and analyzed bound proteins by proteomics. Among other proteins, we identified Vps51 (also referred to as Ang 2 in humans), which is known to be part of the multi-subunit GARP/EARP tethering complexes[29,30]. The association between Stx6 and Vps51 was confirmed by co-immunoprecipitation (Supplementary Fig. 3a) and by binding of Vps51 to liposomes containing full-length Stx6 but not ΔN-Stx6 (Fig. 4b). In addition, upon co-transfection of GFP-Vps51 and mRFP-Stx6, the proteins not only colocalized in the TGN as reported[29] but also on small vesicles in cytoplasm (~54% of Stx6 vesicles; Fig. 4c, data not shown). These data agree with previous observations showing that Vps51 forms a specific complex with the N-terminal domain of Stx6 (ref. [22]).

Is Vps51 both necessary and sufficient for the targeting of injected Stx6 liposomes? To this end, we established stable Vps51 knockdown cell lines (Fig. 4d), and measured targeting of injected Stx6 liposomes. Indeed, co-localization of these liposomes with M6PR-containing vesicles was significantly decreased (Fig. 4d). Next, we fused the C-terminal domain of the outer mitochondrial protein monoamine oxidase (MAO) to a construct containing Vps51 and GFP (Vps51-GFP-MAO; Supplementary Fig. 3b). This fusion protein was efficiently targeted to mitochondria (Supplementary Fig. 3c). Furthermore, Vps51-GFP-MAO co-precipitated with Vps52, suggesting that the fusion protein is still capable of forming GARP complexes (Supplementary Fig. 3d). Injection of Stx6 liposomes resulted in the gradual accumulation of the proteoliposomes (PLs) on the surface of Vps51-GFP-positive mitochondria (Fig. 4e and Supplementary Fig. 3e, 3f). No such accumulation was observed when protein-free liposomes were used or when Vps51 was omitted (Fig. 4e and Supplementary Fig. 3e, f). These data indicate that Vps51 is sufficient for orchestrating the capture of injected Stx6 liposomes. On the other hand, no intermixing between the GFP-labeled outer mitochondrial membrane and the Rh-PE-labeled liposome membrane was observed. Thus, Stx6, while being sufficient for targeting, is not able to mediate fusion with mitochondria. Furthermore, we observed that in transfected cells, endogenous Stx6- or M6PR-containing endosomes selectively accumulated around the labeled mitochondria, whereas no such accumulation was seen for Tfn- or Golgin97-positive vesicles (Fig. 4f). Moreover, no accumulation was observed in cells expressing GFP-MAO (Supplementary Fig. 3g) confirming that Vps51 is both necessary and sufficient for mitochondrial mistargeting of both injected Stx6 liposomes as well as endogenous Stx6/M6PR-containing trafficking vesicles.

**Vps13B is recruited to Stx6- and Stx13-containing vesicles.** As shown above, liposomes containing both Stx6 and Stx13 differ from Stx6 liposomes by being targeted to Tfn-positive endosomes as well as to M6PR-positive vesicles. This is surprising since

**The N-terminal domain of Stx6 mediates targeting by Vps51.** To clarify whether the N-terminal domain of Stx6 is required for targeting of the liposomes to the M6PR compartment, we injected liposomes containing an N-terminally truncated mutant of Stx6 (ΔN-Stx6, residues 169–255). No co-localization with any organelle markers was observable (Fig. 4a). Next, we incubated

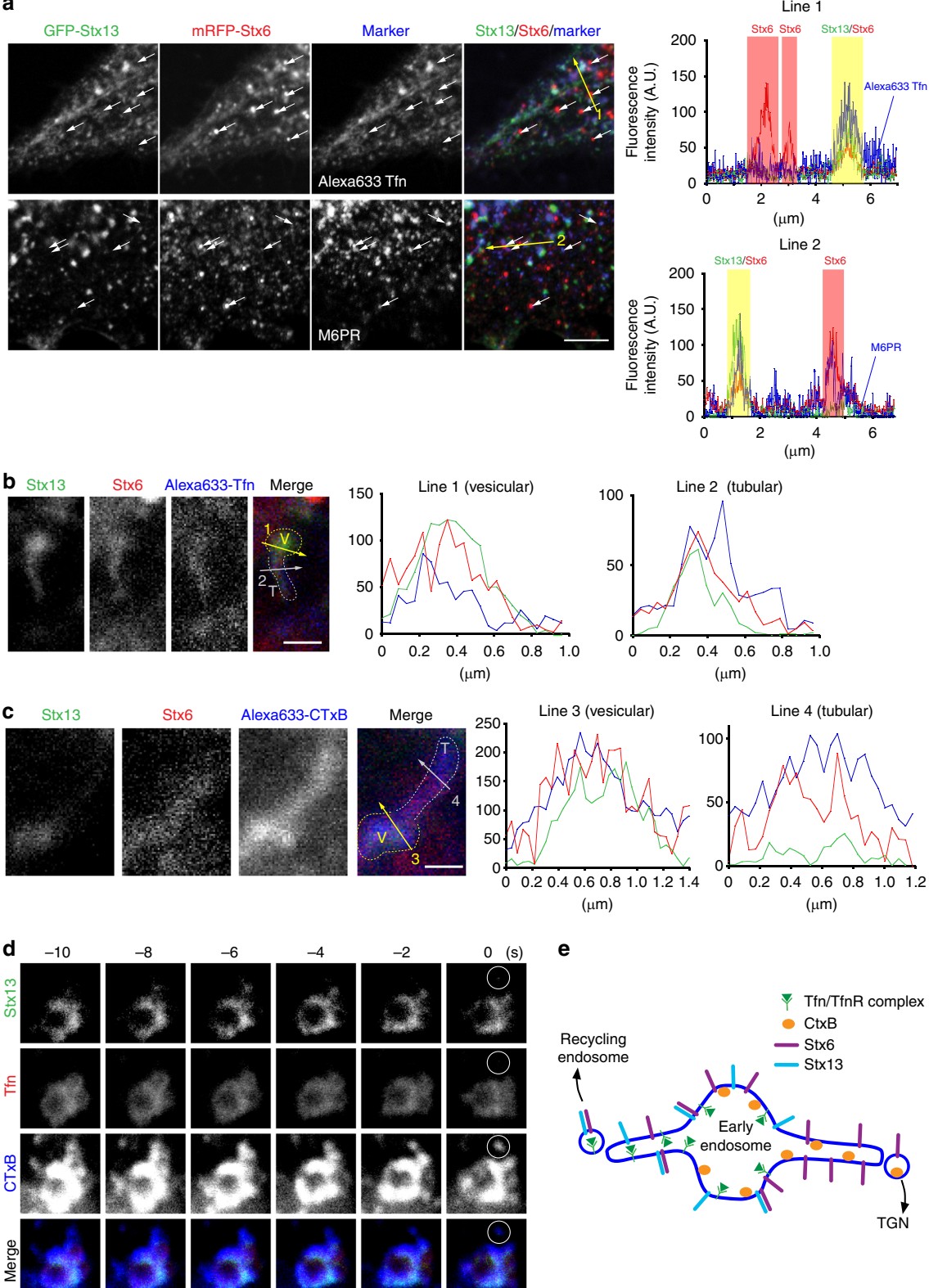

liposomes containing only Stx13 are not specifically targeted (Fig. 2c). Moreover, almost all of the endogenous endosomes involved in Tfn recycling contain both SNAREs. These data imply that recycling endosomes recruit tethering proteins that recognize both Stx6 and Stx13. The only known protein suggested to bind to both SNAREs on early endosomes is EEA1[31,32]. Since the presence of additional hitherto unknown tethering factors cannot

be excluded, we used SNARE-containing liposomes to search for cytoplasmic proteins specifically binding to both SNAREs. Among others, we detected Vps13B, one of four Vps13 paralogs in mammals that is also referred to as COH1 due to its association with Cohen syndrome, a severe inheritable disease[33] (Supplementary Table 1). Analysis by immunoblotting confirmed the presence of Vps13B on liposomes containing both SNAREs

**Fig. 3** SNAREs are selected during budding of transport vesicles. **a** Association of Stx6 and Stx13 with transport vesicles containing either Tfn or M6PR. Endosomal compartments in cells expressing mRFP-Stx6 and GFP-Stx13 were labeled either by internalization of Alexa Fluor 633-Tfn or by immunostaining for anti-M6PR antibody. White arrows indicate Stx6(+)/Stx13(−) vesicles. Most of them are negative for Tfn (arrows, top), but several of them are positive for M6PR (arrows, bottom). Scale bar, 5 μm. Right: intensity plots of the line scans shown by yellow lines in the left images. Stx6(+)/Stx13(+) vesicles (shadowed by light yellow) co-localize with both Tfn (top) and M6PR (bottom), but Stx6(+)/Stx13(−) vesicles are negative for Tfn and positive for M6PR. **b**, **c** Snapshots of tubular structures (T) extending from vesicular endosomes (V) from a movie. **b** Both domains are triple positive for Stx13, Stx6, and Alexa633-Tfn. Intensity plots of the line scans are shown on the right side. **c** A tubule is shown that contains the retrograde marker Alexa647-cholera toxin B (CTxB) and that is positive for Stx6 but negative for Stx13. Right panels show line scans shown by lines in the left panels. **d** Snapshots from a movie showing budding of a CTxB-positive vesicle (white circles in the last frame) from a globular endosome that is positive for CTxB but negative for Stx13 and Tfn. Scale bar, 1 μm. **e** Summary showing sorting of cargos and SNARE proteins in early endosomes

---

whereas no or only reduced binding was observable to liposomes containing either of the SNAREs (Fig. 5a). Note that using this assay, we were unable to detect an interaction with EEA1 (Fig. 5a).

Human Vps13B is a large protein of 3997 amino acids (NP_689777). *VPS13* was originally identified in yeast but is now known to be conserved in eukaryotes[34]. In yeast, Vps13p is widely distributed in the endosomal system and has been invoked both in membrane traffic and in the formation of membrane contact sites[34]. The human *Vps13* gene family contains four members (*Vps13A-D*) that appear to have functionally diversified[35]. Recently, Vps13B was shown to be primarily localized to the Golgi apparatus and to be an effector of Rab6, suggesting a function in the organization of Golgi membranes[36].

To confirm the interaction between Stx6 and Stx13 and Vps13B, we immunoprecipitated endogenous Vps13B from detergent extracts of HeLa cells. Co-precipitation with both Stx6 and Stx13 but not with VAMP4, Stx4 or Stx16 was observed (Fig. 5b). Conversely, Vps13B co-precipitated with Stx6 and Stx13 but not with the other tested SNAREs (Fig. 5c). To test whether the N-terminal domain of Stx13 is required for Vps13B-binding, we transfected cells, together with FLAG-tagged Vps13B with (i) GFP-tagged Stx13, (ii) Stx13 with the N-terminal domain being truncated (GFP-ΔN-Stx13), and (iii) a chimera in which the N-terminal domain of Stx13 was replaced with that of Stx6 (GFP-Stx13–6). FLAG-Vps13B was then immunoprecipitated using an anti-FLAG antibody. Full-length GFP-Stx13 and GFP-Stx13–6 but notΔN-Stx13 co-precipitated, suggesting that the both N-terminal domains are required whereas the nature of the SNARE motif is probably not relevant (Fig. 5d).

In HeLa cells, Vps13B was previously shown by immunocytochemistry to be concentrated as a peripheral membrane protein of the Golgi apparatus, but the authors also noted that the protein has a more widespread distribution that was not confined to the Golgi[37]. Immunolabeling for endogenous Vps13B confirmed these findings (Fig. 5e), with very similar patterns being observable when a Vps13B-GFP and FLAG-Vps13B construct was expressed (see Fig. 6c and Supplementary Fig. 5b, data not shown). To examine whether Vps13B is associated with Stx13 and Stx6 containing transport vesicles in the endosomal region, we focused on the peripheral region of the cell, again using expression of tagged SNAREs to facilitate localization. As shown in Fig. 5e, f, endogenous Vps13B displayed co-localization with vesicles positive for both mRFP-Stx6 and GFP-Stx13 (34%) whereas co-localization with vesicles containing either mRFP-Stx6 or GFP-Stx13 was low (7% and 10%, respectively). Furthermore, partial (~40%) colocalization was also found with TfnR both in the cell periphery (Fig. 5g). In contrast, overlap of Vps13B with endogenous EEA1 and M6PR, respectively, was lower than TfnR (Fig. 5g), and no overlap was observable with the late endosomal markers CD63 and GFP-Rab7, with the mitochondrial marker mitofilin, or with the autophagosomal marker LC3, regardless of whether autophagocytosis was induced by starvation (Supplementary Fig. 4a). Together, these findings

suggest that in the cell periphery, Vps13B selectively associates with recycling endosomes carrying Stx6 and Stx13, in addition to its abundant presence in the "perinuclear cloud" including recycling and late endosomes, lysosomes, and the vesicles of the TGN[38].

**Vps13B binds to specific Rab proteins and phosphoinositides.** Most of the established tethering factors operate by coincidence detection, i.e. they simultaneously bind to zip code molecules such as specific small GTPases including Rabs, Arfs, and phosphoinositides to restrict their distribution and regulate the function[6,7]. When analyzing Vps13B immunoprecipitates by proteomics, both Rab6 and Rab14, which regulate Tfn recycling pathway prior to Rab11 and after Rab5 and Rab4 (ref. [39]), were detected. For further exploration, we expressed tagged variants of the endosomal GTPases Rab4, Rab5, Rab6, Rab11, and Rab14 in HeLa cells, followed by immunoprecipitation of the Rabs and immunoblotting for endogenous Vps13B. A specific interaction was observed for Rab6 (confirming ref. [36]) and for Rab14. None of the other Rabs were detected (Fig. 6a). Similarly, endogenous Rab14 and Rab6 coprecipitated with Vps13B. This was not the case for Rab 11 or Rab5 (Fig. 6b). The interaction of Vps13B with Rab14 was not changed regardless of whether the constitutively active (Rab14(Q70L)) or dominant negative (Rab14(S25N)) form was expressed (Fig. 6c). We then asked to which extent Vps13B-containing vesicles also contain Rab14 and Rab6. Using expression of mRFP-tagged Rab14, we observed a rather widespread distribution in the endosomal system, including the perinuclear cloud (Fig. 6d), in agreement with previous observations[40]. Coexpression of GFP-tagged Vps13B revealed that half of Rab14-positive small puncta in the peripheral cytoplasm were positive for Vps13B (Fig. 6d). In contrast, GFP-Rab6 was almost exclusively localized to structures resembling the TGN, overlapping with endogenous Vps13B (36). These structures, however, displayed a more punctate pattern associated with Rab6-positive TGN structures (Fig. 6d insert).

Since yeast Vps13 has recently been shown to specifically bind to PtdIns(3)P[41], we asked whether mammalian Vps13B displays a similar phosphoinositide binding preference. First, using a cell-free extract of cells expressing Vps13B-GFP, a PIP strip assay was carried out, and bound Vps13B was detected with a GFP antibody. As shown in Fig. 6e, specific binding to PtdIns(3)P but not to any other phosphoinositide was observed. Second, we used beads coated with anti-GFP antibody to isolate Vps13B-GFP, followed by washing and incubation with liposomes. Binding was only observed when the liposomes contained PtdIns(3)P (Fig. 6f). These data suggest that Vps13B specifically binds to PtdIns(3)P-containing membranes although a contribution of other endogenous proteins binding to Vps13B cannot be completely ruled out (Fig. 6f).

**Vps13B functions as a tethering protein.** The data above show that Vps13B specifically interacts with Stx6, Stx13, and with

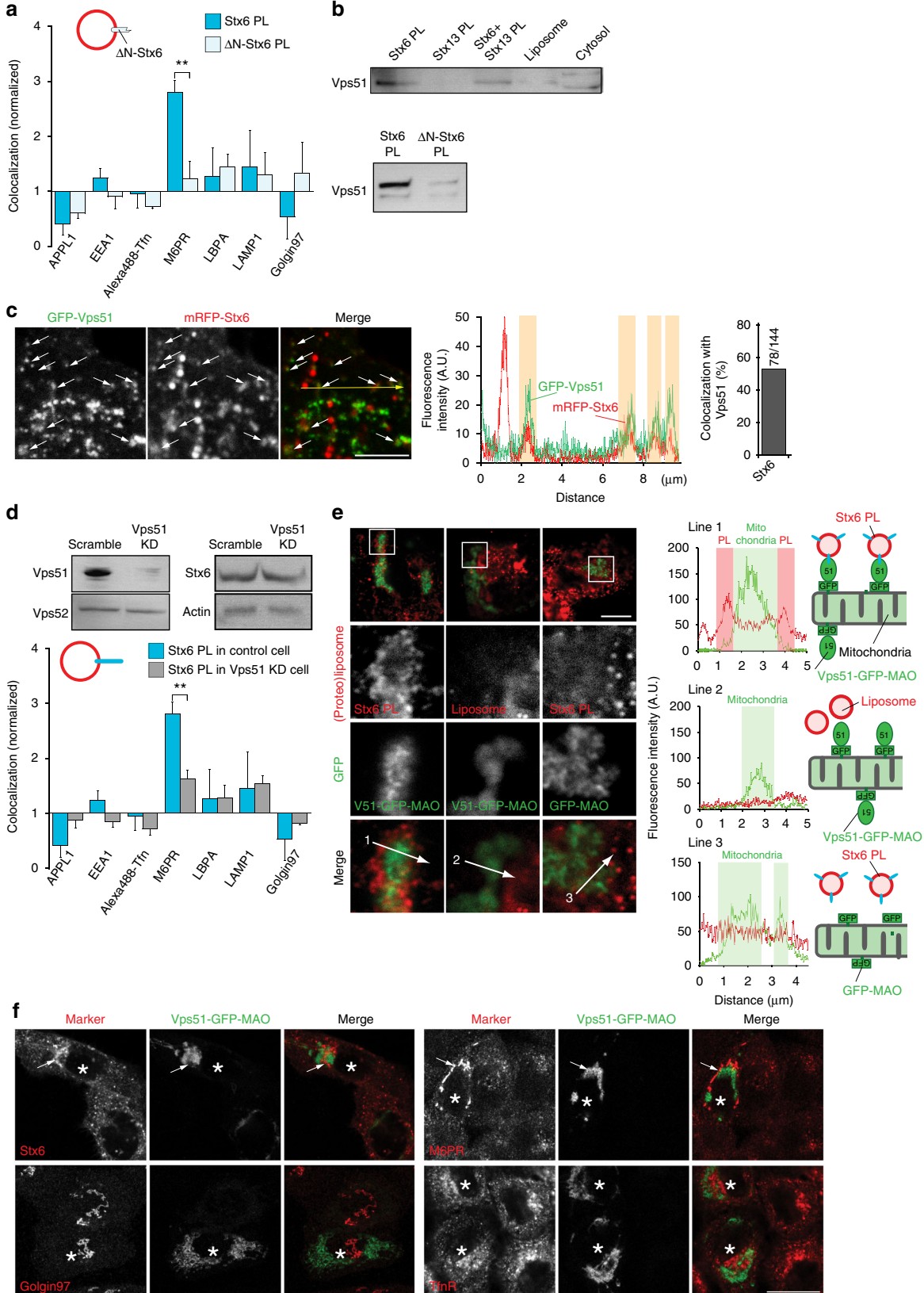

PtdIns(3)P, and they confirm and extend the notion that Vps13B specifically interact with select Rab GTPases—all features typical for bona fide tethering factors. In the final set of experiments, we therefore asked whether Vps13B does indeed function as tethering factor that is responsible for the targeting of Stx6+/Stx13+

liposomes to TfnR-containing compartments. To this end, we targeted Vps13B to mitochondria by expressing a Vps13B-GFP-MAO construct (Supplementary Fig. 5a), thus using the same strategy as for Vps51 above, resulting in mitochondrial localization of the fusion protein (Supplementary Fig. 5b). Due to low

**Fig. 4** The N-terminal domain of Stx6 mediates targeting via Vps51. **a** Deletion of the N-terminus of syntaxin 6 (169–255) (ΔN-Stx6 PL) results in loss of specific targeting. For comparison, the results obtained for liposomes containing full-length Stx6 are shown. **P < 0.01, all determined by unpaired *t*-test. **b** Recruitment of Vps51 to liposomes containing Stx6, Stx13, Stx6, and Stx13 together (top panel) and N-terminally truncated Stx6 (bottom), with protein-free liposomes serving as control. A cell-free supernatant obtained from HeLa cells was incubated with the respective liposomes, followed by separation of unbound protein by a flotation gradient. The liposome fraction was analyzed by immunoblotting for Vps51. **c** Peripheral region of a cell expressing GFP-Vps51 and mRFP-Stx6. White arrows indicate Vps51- and Stx6-double positive vesicles. The intensity plot of the yellow line in the left image shows co-localization of Vps51 and Stx6. A bar graph shows the percentage of Stx6-positive vesicles that are also positive for Vps51. Vesicles were counted from four images, obtained in three independent experiments. The number of Vps51-positive and total vesicles counted in these experiments are shown. Scale bar, 5 μm. **d** Knockdown (KD) of Vps51 in HeLa cells reduces targeting specificity of liposomes containing Stx6. Top panels show immunoblots of marker proteins in knockdown cell lines (scramble: siRNA with scrambled sequence). **P < 0.01, all determined by unpaired *t*-test. **e** Re-routing of Stx6 PL to mitochondria expressing Vps51-GFP, measured 60 min after injection. Vps51-GFP was targeted to the outer membrane of mitochondria by fusing it to the mitochondria-targeting signal of monoamine oxidase (MAO). No mitochondrial association was found when protein-free liposomes were injected or when the Vps51 was omitted. Scale bar, 7.5 μm. On the right are the line scans of the white arrows in the bottom panels as indicated. **f** Accumulation of endogenous vesicles containing Stx6 or M6PR around mitochondria upon expression of Vps51-GFP-MAO. Note that such accumulation was not observable for vesicles containing TfnR or Golgin 97 (bottom panels). Scale bar, 20 μm

expression levels and low efficiency of transfection, experiments involving the injection of liposomes were not feasible. Instead, we tested whether endogenous endosomes were re-distributed. Intriguingly, TfnR-positive endosomes became scattered (Fig. 7a), associated with a dramatic reduction of the perinuclear index (Fig. 7a) that was not seen for organelles containing M6PR (Fig. 7a) or Golgin97 (Supplementary Fig. 5c). At higher magnification, some TfnR-positive endosomes seemed to distribute along mitochondria (Fig. 7a). Next, we generated stable cell lines expressing Vps13B shRNA in HeLa cells, resulting in significant downregulation of the protein (Fig. 7b and Supplementary Fig. 5d, e). We then injected Stx6/Stx13 liposomes. Targeting of these liposomes to Tfn-positive endosomes was significantly decreased, whereas targeting to the M6PR-positive compartment was not affected (Fig. 7c). These results suggest that Vps13B operates as a tethering factor in the Tfn recycling pathway, which is governed by a combination of the Q-SNARE proteins Stx6 and Stx13.

The Tfn/TfnR complex is known to be recycled to the plasma membrane by two different pathways: (i) a Rab4 and Rab35-dependent recycling pathway (fast recycling) in which vesicles formed at the early endosomes in the cell periphery are directly targeted to the plasma membrane, and (ii) a Rab11-dependent pathway (slow recycling) in which tubulo-vesicular transport organelles are first delivered to the perinuclear recycling endosome from where they are returned to the plasma membrane (Fig. 7g)[42]. To examine at which step Vps13B operates, we monitored the fate of endocytosed Alexa Fluor 568-Tfn over time in wt and Vps13B KD cells using time-lapse microscopy. While neither the rate of Tfn clearance (Fig. 7d) nor the surface levels of M6PR and TfnR (Supplementary Fig. 6a) were changed significantly, we noted a conspicuous loss of perinuclear accumulation in Vps13B KD cells, particularly evident 15–30 min after the pulse (Fig. 7d). A similar re-distribution toward the cell periphery was seen for Rab11 and Rab14, both specific for the late recycling pathway, whereas markers for early endosomes (EEA1), late endosomes (CD63), retrograde pathway (M6PR), and mitochondria (Mitofusin2) remained unchanged (Supplementary Fig. 6b). The TGN (Golgin97) was weakly fragmented, as previous reported[36].

These data suggest that Vps13B may act as a tethering factor in the trafficking step between early and recycling endosomes. To confirm this interpretation, we immunoisolated TfnR containing vesicles from cell extracts with anti-TfnR antibody and analyzed them by immunoblotting for the presence of endosomal and lysosomal markers. As shown in Fig. 7e, Vps13B knockdown resulted in a reduction of co-distribution of TfnR with Rab11, an increase in Rab35 and Rab4, and also an increased presence of LAMP1. Moreover, the overall protein levels of TfnR were

reduced in Vps13B KD cells, which was prevented by addition of chloroquine, a lysosomal inhibitor (Fig. 7f). Thus, knockdown of Vps13B results in a selective inhibition of the slow TfnR recycling pathway, which is compensated by an increase in the fast recycling pathway, and probably also an increase mis-targeting of TfnR-positive vesicles to lysosomes (see cartoon in Fig. 7g).

## Discussion

Using artificial vesicles with a minimalistic composition as tools, we have shown that SNAREs alone, in the absence of Rab proteins or phosphoinositides, are capable of directing trafficking vesicles to the correct target compartment in the endosomal pathway. Targeting is surprisingly accurate even though it does not appear to be as precise as that of endogenous trafficking organelles. Moreover, our data suggest that targeting information is encoded in the N-terminal domains of SNAREs instead of in the SNARE motifs. The N-terminal domains are known to be critical for the recruitment of additional proteins. Most prominently, such proteins include Sec1/Munc18 (SM) and CATCHR (complex associated with tethering containing helical rods) proteins that are required for regulating SNARE assembly during fusion. Similarly, complexes between individual tethering factors and individual SNAREs were previously reported (see Introduction). However, these interactions are so far mainly thought to ensure that SNAREs or SNARE acceptor complexes are present and ready for fusion at the site where tethering occurs[6,20].

Our findings document that SNARE proteins may play a much more important role in vesicle targeting than previously appreciated. They highlight a second function of some of the SNAREs that becomes relevant immediately after vesicle budding and that is independent of (although connected with) their established role in membrane fusion. The emerging picture shows further that it is not only the individual SNARE but rather the specific combination of SNAREs that, at least in the examples studied here, decides about the recruitment of tethering factors and thus the destination of the vesicles. Such combinatorial coding may explain how specificity can be achieved despite the broad distribution of individual SNAREs, particularly in the endocytotic limb of the secretory pathway[5]. Indeed, only a fraction of the dozen or more SNAREs operating in the endocytotic pathway would be needed for creating a specific SNARE combination for each trafficking step, even if constraints apply for meeting the "QabcR-rule" for functional SNARE complexes[14]. Differential sorting of SNAREs during the budding of a trafficking compartment is thus a critical determinant of vesicle targeting whereas less specificity may be involved in SNARE-pairing required for the final fusion step. This also agrees with previous findings showing that in vitro, tethering/docking of early endosomes, in contrast to fusion, is

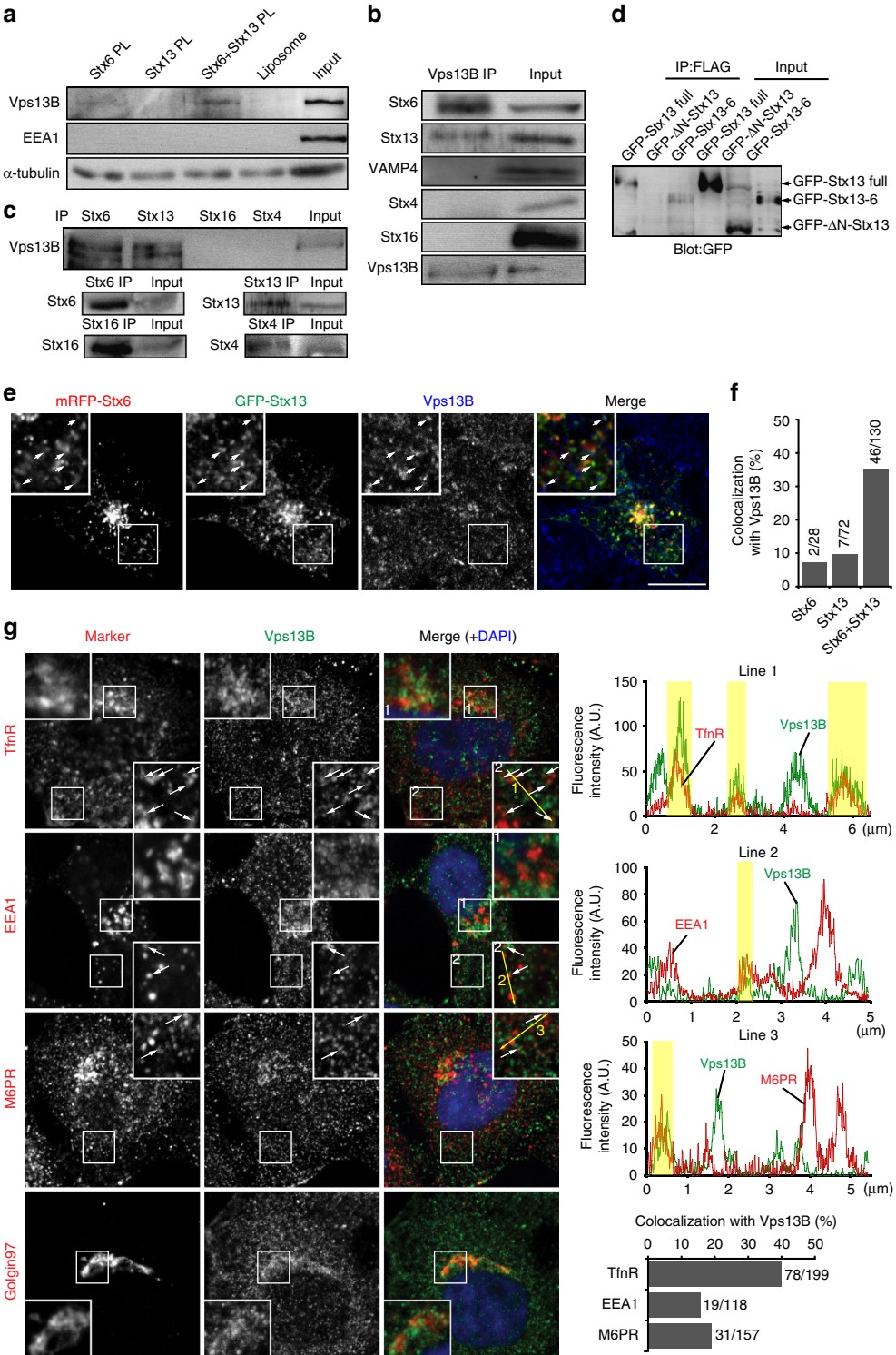

**Fig. 5** Vps13B binds to Stx6- and Stx13-positive vesicles. **a** Recruitment of Vps13B and EEA1 to liposomes containing Stx6, Stx13, Stx6, and Stx13 together. For details, see legend to Fig. 4b. α-Tubulin was used as loading control, because the protein binds to liposomes dependent on the lipid composition (ref. [23]). **b**, **c** Co-immunoprecipitation of Vps13B and SNARE proteins; **b** shows co-precipitation of Stx6 and 13 but not of any other SNARE with Vps13B. **c** Conversely, Vps13B co-precipitates with Stx6 and Stx13 but not with Stx16 and Stx4. The bottom panel shows that all SNAREs were efficiently immunoprecipitated. **d** The N-terminal domain of Stx13 is necessary for binding to Vps13B. Lysates from cells expressing Vps13B-FLAG and GFP-Stx13, GFP-ΔN-Stx13, or GFP-Stx13-Stx6 chimera were immunoprecipitated with anti-FLAG antibody and probed by immunoblotting for GFP. **e** Immunofluorescence for Vps13B of cells expressing mRFP-Stx6 and GFP-Stx13. Arrows in the insert indicate triple-positive vesicles. Scale bar, 10 μm. **f** Co-localization between Stx6, Stx13, and Vps13B in the peripheral region of the imaged HeLa cells. Vesicles were counted from nine images, obtained in three independent experiments, with the degree of overlap shown in the bar graph on the right (numbers denote total number of vesicles analyzed). **g** Co-localization of Vps13B with TfnR, EEA1, M6PR, and Golgin97. White arrows show co-localized vesicles. Scale bar, 10 μm. Intensity plots of exemplary line scans are shown on the right. A bar graph shows the percentage of TfnR-, EEA1-, M6PR-positive vesicles, which are also positive for Vps13B

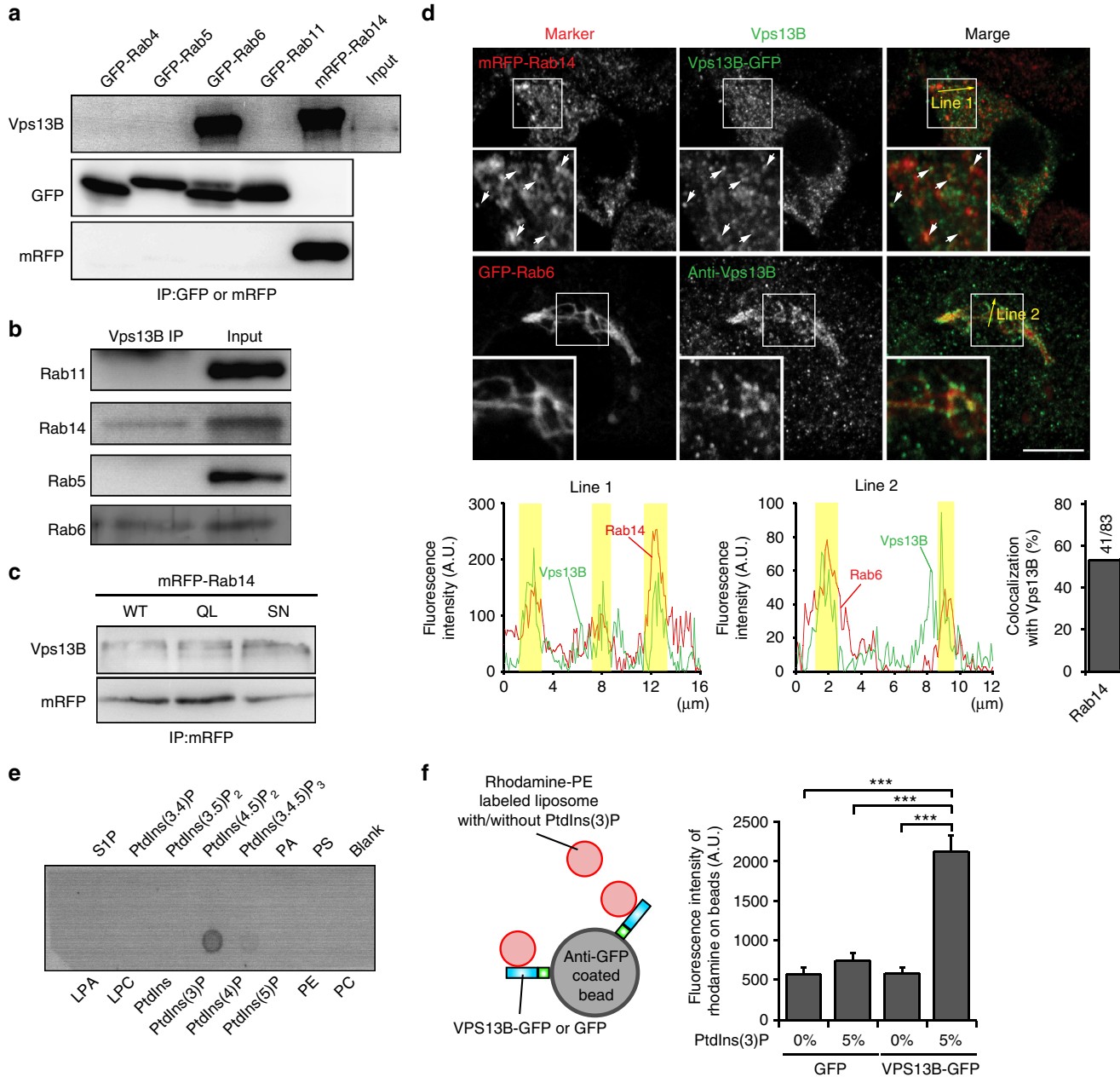

**Fig. 6** Vps13B specifically binds to Rab14 and Rab6 and to PtdIns(3)P. **a**, **b** Association of endosomal Rab proteins with Vps13B, revealed by co-immunoprecipitation of endogenous Vps13B with GFP (or mRFP)-tagged Rab proteins. **a** GFP (or mRFP)-tagged proteins were precipitated from cell lysates using a GFP-trap or anti-mRFP antibody. Bound proteins were analyzed by immunoblotting for Vps13B, GFP, and mRFP. **b** Cross-linked cells were lysed and immunoprecipitated with anti-Vps13B antibody. The interaction of Rab proteins was detected and indicated Rab protein antibodies. **c** Coimmunoprecipitation of VPS13B with mRFP-Rab14-WT, GTP-locked mRFP-Rab14Q70L (QL), or GDP-locked mRFP-Rab14S25N (SN) expressed in HeLa cells. Anti-RFP antibody was used for immunoprecipitation of mRFP-tagged Rab14 proteins. There was no significant difference in the binding of Vps13B to the Rab14 variants. **d** Vps13B co-localizes with Rab6 and Rab14 upon expression in HeLa cells. Top panel: cells were transfected with mRFP-Rab14 and Vps13B-GFP and analyzed for the presence of GFP and mRFP, respectively, by fluorescence microscopy. Bottom panel: cells were transfected with GFP-Rab6. In this case, the distribution of endogenous Vps13B was monitored by immunocytochemistry. White arrows in the upper inserts show vesicles positive for both Vps13B and Rab14. Bottom panels show line scans on the lines in the upper panels and the percentage of colocalization of Rab14 with Vps13B. Scale bar, 10 μm. **e** PIP strip-binding assay of Vps13B. Perinuclear supernatant from Vps13B-GFP overexpressing cells was incubated with a membrane strip containing the phosphoinositides as indicated and probed for bound Vps13B using an anti-GFP antibody. **f** Binding of Rhodamine-labeled liposomes to Vps13B-GFP that was immobilized on magnetic Dynabeads. After incubation, the beads were washed, and bound liposomes were detected by Rhodamine fluorescence. Binding was only detectable when Vps13B-GFP was bound to the beads and when the liposomes contained 5% PtdIns(3)P. Error bars indicate S.E.M., ***$P < 0.001$, determined by unpaired $t$-test

independent of SNARE pairing[43]. In a way it makes sense to select the correct SNAREs during budding rather than having the vesicle shipped to the right place and then, in the end, a decision needs to be made whether fusion is allowed to proceed or aborted

due to lack of pairing specificity. Sorting of SNAREs appears to be mediated by specific interactions with coat proteins such as cla-thrin adaptors[44–46]. For instance, Stx13 is known to interact with the BLOC-1 complex, which controls the formation of recycling

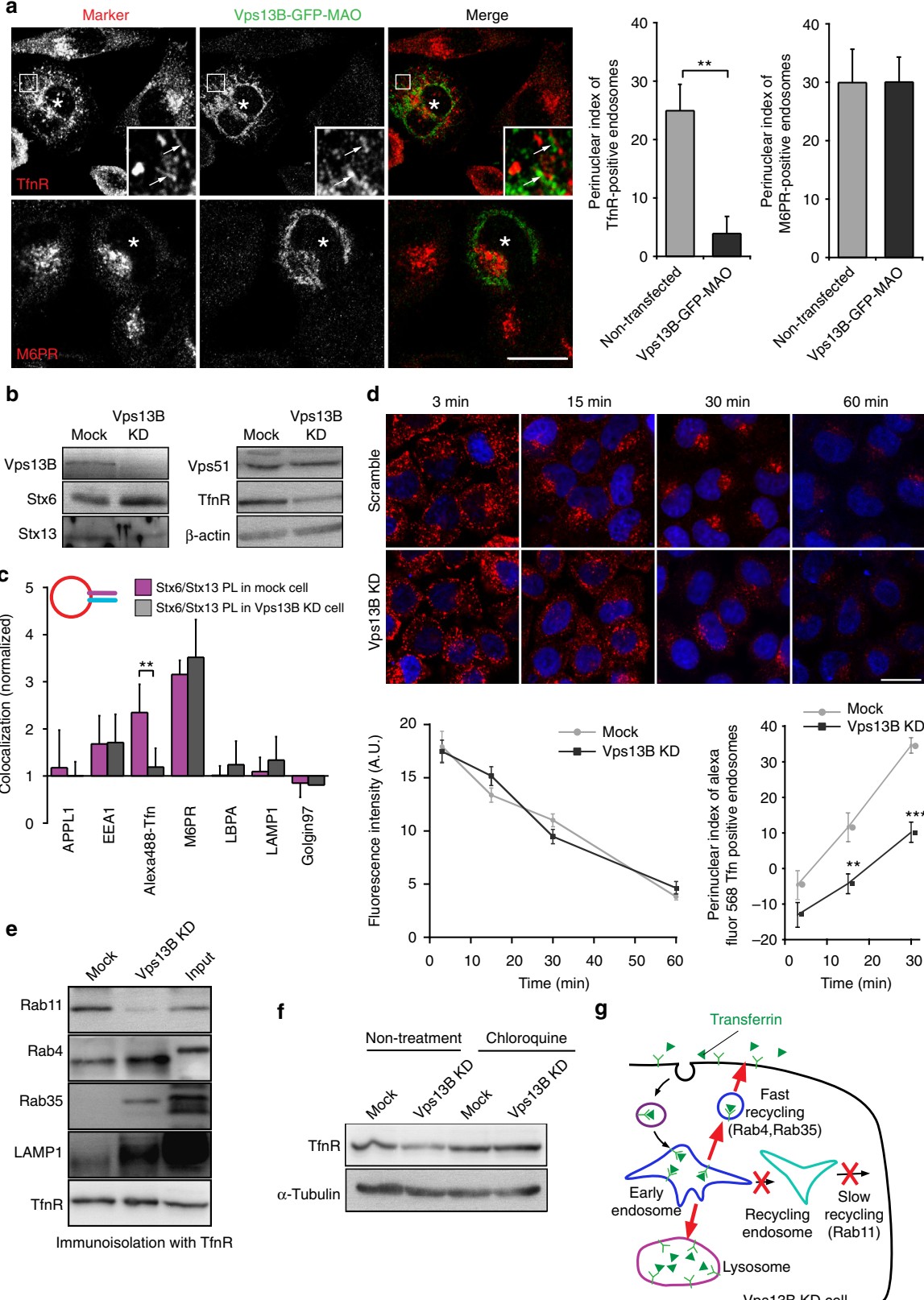

endosomal tubules[47]. However, more work is required for understanding how exactly individual SNAREs are selected or excluded during the formation of a trafficking vesicle from a common precursor membrane, and to establish whether SNARE-mediated targeting by this mechanism also occurs in other domains of the secretory pathway.

While the injection of SNARE-containing, but otherwise "naïve", vesicles revealed sufficiency of SNAREs for targeting and fusion, the SNARE-encoded targeting signals are embedded in complex regulatory networks in which Rab switching and PtdIns phosphates play major roles[15]. Functional tethering factors usually interact with all three of these signal classes, either directly

**Fig. 7** Vps13B functions as a tethering protein for Tfn-positive endosomes. **a** Expression of Vps13B-GFP on the outer mitochondrial membrane by fusing it to a monoaminoxidase fragment (see also Fig. 4e) results in scattering and accumulation of TfnR-containing vesicles around mitochondria (arrows in insert). GFP signal was enhanced by immunostaining with anti-GFP antibody. Right: perinuclear index of TfnR and M6PR distribution upon expression of Vps13B-GFP-MAO. Error bars indicate S.E.M., **$P < 0.01$, determined by unpaired $t$-test. **b** Expression levels of Vps13B and marker proteins in a HeLa cell line stably expressing Vps13B shRNA expressing cell line. The figure shows immunoblots of cell lysates for the proteins indicated. **c** Knockdown of Vps13B selectively reduces targeting of liposomes containing Stx6/Stx13 to Tfn-positive endosomes. **d** Vps13B KD cells, and cells expressing a control construct (scramble) were allowed to internalize Alexa Fluor 568-Tfn for 10 min and chased for indicated time at 37 °C. The images show the change in the distribution of labeled organelles after counterstaining of the nuclei with DAPI (blue channel). Lower left: quantification of fluorescence intensity of Alexa Fluor 568-Tfn over time after pulse-chase of labeled Tfn. The data show mean values ± S.E.M. of three independent experiments. Lower right: quantification of the distribution of Alexa Fluor 568-Tfn-positive endosomes between the peripheral region and the perinuclear region over time. The perinuclear index decreased in Vps13B KD cells, suggesting that transport of Tfn-positive endosomes to the perinuclear recycling endosomes was impaired. Error bars indicate S.E.M., **$P < 0.01$, ***$P < 0.001$, determined by unpaired $t$-test from three independent experiments. **e** Immunoisolation of vesicles containing TfnR reveals a loss of association with Rab11, whereas association with Rab4 and Rab35 was increased in Vps13B KD cells. Moreover, a higher level of LAMP1 was detectable. The figure shows immunoblots for the proteins indicated, with all lanes containing equal amounts of TfnR. **f** Immunoblots of control (mock) and Vps13B-KD cell extracts, normalized to tubulin, reveal that Vps13B-KD leads to a reduction of Tfn levels that is preventable by the inhibition of lysosomal function by 10 μM chloroquine. **g** Cartoon showing the recycling of transferrin in Vps13B KD cells. After delivery to early endosomes by clathrin-mediated endocytosis, it can recycle either directly (fast recycling) using a pathway governed by Rab4 and Rab35, or via recycling endosomes, which is governed by Rab11 in control cells. Vps13B KD appears to block the slow pathway, leading to an increase in the fast recycling route and to an increased degradation via the late-endosome lysosome route

or via intermediate coat proteins, and thus it is probably the combination of all signals that fine-tunes the destination of a given vesicle[48]. However, with few exceptions[49–51], it is not known how exactly these three signal types interact with each other and how the hierarchy between them (if any) is structured. We hope that the approach used here will allow for individual testing of each of the components, thus providing an experimental approach toward addressing these unresolved issues.

Vps51 is part of the GARP complex[29] and the more recently identified and structurally related EARP complex (refs. [30,51]). The GARP complex is required for the return of M6PR from endosomes to the TGN[52]. In yeast, GARP is known to be an effector of the RabGTPase Rab6 and Arl1 (ref. [53]). Fusion at the TGN is thought to involve the SNAREs Stx6, vti1a, and VAMP4, but, intriguingly, syntaxin 16 instead of Stx13 (ref. [54]). In this context, our data revealing that in the absence of Stx13, Stx6 liposomes, regardless of whether vti1a is present or not, are targeted to M6PR-positive compartments can be easily interpreted as Stx6, via binding to the GARP complex by interacting with Vps51, being decisive for targeting. Vps51 is both necessary and sufficient for governing targeting since not only injected Stx6 liposomes but also endogenous M6PR vesicles are mis-directed to mitochondria upon expression of a Vps51-MAO construct. Intriguingly, the EARP complex that also contains Vps51 regulates endosomal recycling[30]. It remains to be established whether both complexes allow for Vps51 binding to Stx6, and if this is the case, how differential targeting of Stx6-containing vesicles is regulated.

*Vps13* was originally found in yeast during classical screens for defects in vacuolar sorting[55]. Apparently, Vps13p is required for trafficking from the yeast endosome to the Golgi complex[56], but it is also involved in endosomal recycling in yeast[57,58]. Furthermore, Vps13p is involved in processes such as sporulation and autophagy, but its mechanism of action is unclear (reviewed in ref. [34]). Surprisingly, in an elegant study, it has recently been shown that Vps13 proteins operate as lipid transfer proteins, capable of transferring phosphatidylserine and phosphatidylethanolamine between liposomal membranes[59]. The activity is mediated by the N-terminal domain whose crystal structure revealed a large hydrophobic cavity capable of binding ~10 glycerolipid molecules[59]. In yeast, Vps13p has been identified at membrane contact sites connecting endoplasmic reticulum, mitochondria, endosomes, and vacuoles[60,61] suggesting that Vps13p mediates lipid transfer at such sites. Interestingly, in mammals, different Vps13 isoforms localize to distinct contacts:

Vps13A is enriched at contact sites between the ER and mitochondria (Vps13A), Vps13C at contacts between the ER and late endosomes/lysosomes, and both isoforms were also found at ER contacts to lipid droplets[59].

Our data now reveal that Vps13B is required for the targeting of TfnR-containing vesicles from early to recycling endosomes. Vps13B shares key properties with other tethering factors such as the ability to bind to Rab proteins, PtdIns phosphates, and SNAREs. Furthermore, EM imaging showed that yeast Vps13p[62] has an elongated seahorse-like structure that is intriguingly similar to the HOPS tethering complex in shape and size[63]. However, HOPS is a multiprotein complex with no significant sequence homology. Thus, Vps13B does not belong to one of the established classes of tethering proteins including long coiled-coil long proteins and multi-subunit complexes[7] and may be the first representative of a novel and third class of tethering proteins.

In summary, Vps13 proteins appear to be "moonlighting" as they are involved both in lipid transfer and tethering. This is fascinating since both functions require a connection between two different membranes before the second step—membrane fusion or lipid transfer—is carried out. It is possible that yeast Vps13p represents an ancient form that embodies both functions, whereas in mammals these functions have diversified in the four related isoforms (Vps13 A–D). Indeed, unlike Vps13A and C, Vps13B is not associated with the ER (Supplementary Fig. 4a) as it lacks the FFAT motif needed for ER association[59]. Instead, Vps13B was previously found to be mainly associated with the Golgi where it was reported to regulate the integrity of Golgi stacks due to its function as an effector of Rab6 (ref. [36,37]). Moreover, the homology of the N-terminal domain of Vps13B with those of Vps13A and Vps13C is lower, but it needs to be tested whether Vps13B can function as lipid transfer protein or not. Functional diversification of the mammalian Vps13 isoforms is also supported by the fact that nonfunctional variants of individual isoforms cause different diseases[34]. Moreover, the protein ATG2A may be an additional, albeit more distantly related member of the mammalian Vps13 family: it has high homologies with Vps13 in the N- and C-terminal regions, exhibits a similar elongated seahorse-like shape, and it mediates ER-phagophore association and/or tethering[64].

In conclusion, we show that distinct sets of SNAREs that are selected during budding of a trafficking vesicle determine targeting by selectively recruiting different tethering factors. This is a new function of SNAREs that is independent of their role in catalyzing fusion and also appears to be independent of the rules

governing SNARE pairing between two membranes in preparation for fusion. Combinatorial coding overcomes the "disadvantage" of the broad distribution of SNAREs. Vps13B is the first example of a tethering factor that requires the simultaneous presence of two SNAREs for recruitment, and it is conceivable that additional tethering factors exist recognizing different SNARE combinations. Our injection of artificial vesicles with a defined composition provides a powerful tool for the identification of tethering mechanisms and for determining the specificity in the vesicle targeting of vesicles.

## Methods

**Materials**. All phospholipids were obtained from Avanti Polar Lipids. Primary antibodies used were obtained from the following companies: anti-APPL1 (3858), anti-LC3B (4599), and anti-Rab7 (9367) from Cell Signaling; anti-EEA1 (612006) and anti-GM130 (560257) from BD Biosciences; anti-M6PR (ab2733), anti-LAMP1 (ab24170), anti-mitofilin (ab110329), anti-mitofusion2 (ab101055), and anti-cathepsin D (ab6313) from Abcam; anti-Tfn receptor (sc-65882) from Santa Cruz Biotechnology; anti-LBPA (Z-PLBPA) from Echelon; anti-RFP (R10367), anti-Golgin97 (A-21270), anti-Vps52 (PA5–24408), and anti-Rab11 (71–5300) from Thermo Fischer Scientific; anti-PDI (700782) from ABfinity; anti-Vps51 (HPA061447), Vps13B (HPA043865) from Atlas antibodies; anti-Rab6 (10187–2-P) and anti-Rab35 (11329–2-AP) from Proteintech; anti-CD63 (H5C6) from Developmental Studies Hybridoma Bank; anti-Rab14 (R0656) from Sigma-Aldrich; anti-GFP (132002), anti-α-tubulin (302211; 1:10,000 dilution for Western blotting), anti-β-actin (251003; 1:10,000 dilution for Western blotting), anti-Rab5 (108011), anti-Stx6 (110062), anti-Stx13 (110132), anti-syntaxin 4 (110042), anti-syntaxin 16 (110162), and anti-VAMP4 (136002) from Synaptic Systems. For Western blotting, the appropriate primary antibodies were used at a dilution of 1:1000 (or as otherwise stated). Alexa Fluor 488-, Cy3-, or Cy5-conjugated goat anti-mouse (115–545–166, 115–165–146, 115–175–166) and goat anti-rabbit IgG (111–545–144, 111–165–144, 111–175–144) were from Jackson ImmunoResarch Laboratories and HRP-conjugated goat anti-mouse (STAR117P) and goat anti-rabbit (5196–2504) were obtained from Bio-Rad. Alexa Fluor 488- (T13342), Alexa Fluor 568- (T23365), or Alexa Fluor 633-conjugated Tfn (T23362) and Alexa Fluor 647-conjugated CTxB (C-34778) were from Molecular Probes; Phalloidin Cruz-Fluor 633 conjugated (sc-363796) was from Santa Cruz Biotechnology. N-ethylmaleimide, and 4′,6-diamidino-2-phenylindole, dihydrochloride (DAPI) (D8417) and chloroquine (C6628) were from Sigma-Aldrich.

**DNA constructs**. Mouse full-length Vps51 cDNA was cloned into pGFP plasmid. For generating mouse full-length Vps13B cDNA constructs fused with GFP, cDNAs were amplified as four different fragments with over 20 bp overlap region each other from cDNA made from mouse liver total RNA, and the fragments and pEGFP plasmid (pEGFP-N1 or pEGFP-C1) were assembled using the HiFi DNA Assembly cloning kit (New England Biolabs) (all the primer sequences used in this study are provided in Supplementary Table 2). For the construction of the Vps51-GFP-MAO or Vps13B-GFP-MAO expression vector, human full-length Vps51 or Vps13B and human monoamine oxidase B (MAO-B) C-terminal TMD (469–520) were cloned into pEGFP-N vector (Clontech). shRNA vectors of Vps51 and Vps13B were from GE Dharmacon. Rat Stx6and rat Stx13 cDNA was subcloned into pEGFP-C vector. All clones were verified by sequencing.

**Purification of proteins**. SNARE proteins (Stx13, vti1a, Stx6, VAMP4, syntaxin 7, vti1b, syntaxin 8, VAMP8 derived from *Rattus norvegicus*) were cloned into pET28 vectors (Merck Millipore). To create N-terminally deleted variants of Stx6 and Stx13, the fragments of Stx6 (residues 169–255) and Stx13 (residues 140–274), respectively, were cloned into pET28a. The proteins were expressed in *Escherichia coli* strain BL21 (DE3) and purified via nickel nitrilotriacetic acid chromatography (Qiagen). The His-tags of all proteins were removed by using thrombin cleavage. All proteins were further purified by ion-exchange chromatography. All proteins were 95% pure, as judged by SDS–PAGE and Coomassie blue staining.

**Preparation of PLs**. To prepare large unilamellar vesicles (LUVs), lipids in chloroform:methanol (2:1) were mixed at a ratio of 79.7% PC (L-α-phosphatidylcholine), 20% PS (L-α-phosphatidylserine) and, 0.3% Rhodamine-PE (1,2-dioleoyl-*sn*-glycero-3-phosphoethanolamine-*N*-lissamine rhodamine B sulfonyl ammonium salt) (molar ratios) and the solvents were evaporated. The dried lipid film was re-dissolved in diethyl ether followed by addition of HB150 buffer (150 mM KCl, 20 mM Hepes (pH 7.5)) containing 1 mM DTT. After dispersal by sonication, the diethyl ether was removed by evaporation. Liposomes were extruded using polycarbonate membranes with a pore size of 100 nm (Avanti Polar lipids) to give uniformly distributed LUVs. Incorporation of proteins into LUVs was achieved by *n*-octyl-β-D-glucoside (OG)-mediated reconstitution. Proteins in OG were mixed with LUVs and then detergent was removed by overnight dialysis at 4 °C in HB150 buffer containing SM-2 biobeads (BioRad). The protein-to-phospholipid molar ratio was adjusted to that of early[23,25]. Accordingly, for EE-

SNARE PL, the protein-to-lipid ratio of Stx13, vti1a, Stx6 6, and Vamp4 was 1:2000, 1:10,000, 1:1200, and 1:15,400, respectively. For PLs containing only two SNAREs, a protein:phospholipid ratio of 1:2000 was used for each SNARE protein, and for PLs containing only one SNARE, the ratio was 1:1000. Finally, 4-LE-SNARE-PL, a ratio of 1:2000 was used for each of the four SNARE proteins.

**Protein recruitment to liposomes**. The preparation of cytosol fractions was carried out as described[23]. Briefly, HeLa cells were homogenized in homogenization buffer (250 mM sucrose, 3 mM imidazole-HCl, pH 7.4) with a protease inhibitor cocktail. Debris and membranes were removed by two consecutive centrifugation steps at 3000×g for 15 min and at 100,000 × g for 60 min, respectively. PLs contained, in addition to the indicated components, 1% biotinyl Cap PE. They were incubated with the cytosol fraction for 30 min at 37 °C. Streptavidin-coupled Dynabeads (Invitrogen) were subsequently added to the mixture and incubated for 30 min at 4 °C. The beads were washed three times with HB150 (150 mM KCl, 20 mM HEPES (pH7.5)). Bound proteins on liposomes were identified by proteomic analysis involving mass spectrometry[65] or by Western blotting.

**Immunostaining**. Injected HeLa cells were fixed with 4% paraformaldehyde (Sigma-Aldrich) in PBS for 10 min and washed with PBS three times. The cells were incubated overnight at 4 °C with primary antibody diluted 1:300–1000 in PBS containing 0.05% saponin and 1% goat serum. The coverslips were then washed three times with PBS and incubated with Cy2- or Cy3-labeled secondary antibodies, diluted at 1:600 in PBS containing 0.05% saponin and 1% goat serum for 90 min at room temperature. After washing with PBS, the coverslips were mounted in Dako Fluorescent Mounting Medium (DakoCytomation). For immunostaining of cell surface proteins, cells were incubated with primary antibody at 4 °C for 60 min and cells were washed with PBS and then fixed with 4% paraformaldehyde in PBS for 10 min. After washing with PBS, the cells were incubated with Cy2- or Cy3-labeled secondary antibodies in PBS containing 0.05% saponin and 1% goat serum for 60 min at room temperature.

**Internalization of Tfn and CTxB**. For measuring internalization and recycling of Tfn, cells were starved for 3 h in Dulbecco's modified Eagle medium (DMEM) containing 0.2% bovine serum albumin and then incubated with 5 μg/ml Alexa Fluor 568- or Alexa Fluor 633-conjugated Tfn at 4 °C for 60 min. After cells were washed with ice-cold PBS, they were incubated in pre-warmed serum containing DMEM at 37 °C for the indicated times. For CtxB internalization, Vero cells were incubated with 10 μg/ml Alexa Fluor 647-conjugated CtxB at 4 °C for 60 min. After being washed, the cells were incubated in pre-warmed injection medium (F12 medium (Invitrogen), supplemented with 10% fetal calf serum (FCS), 10 mM HEPES (pH7.5), and 100 units/ml each of penicillin and streptomycin) and observed by confocal microscopy.

**Cell culture and preparation of early/late endosomes**. HeLa cells and Vero cells were grown in DMEM (Lenza GmbH) with the following additions: 10% FCS (PAA laboratories GmbH), 4 mM glutamine (Lenza GmbH), and 100 units/ml each of penicillin and streptomycin (Lenza GmbH). Plasmid DNAs were transfected into HeLa cells using Lipofectamine LTX reagents according to the manufacturer's instructions (Thermo Scientific).

For the generation of knockdown cell lines for Vps51 and Vps13B, five different target shRNA-containing vectors (GE Dharmacon) were used for transfection, followed by the selection of stable cell lines using 1 μg/ml puromycin. Screening was performed by Western blotting.

For the generation of a cell line stably expressing GFP-Rab7, the plasmid was transfected with Lipofectamine 2000 (Invitrogen), and stably expressing cells were selected using 600 μg/ml G418-containing medium. Among several clones, a line was selected in which the expression levels and their distribution of GFP-Rab7 in the cells were comparable to those of endogenous Rab7.

Early endosomes were prepared as in Koike and Jahn[23]. For the isolation of late endosomes, GFP-Rab7 stably expressing HeLa cells were used. The cells were harvested by trypsin/EDTA treatment (Lonza GmbH), followed by washing once with fresh culture medium and once with internalization medium (OptiMEM, containing 10 mM glucose; Invitrogen). The cellular pellets were resuspended in homogenization buffer with protease inhibitor (Complete EDTA-free (Roche)) and cracked using a ball homogenizer with a clearance of 0.02 mm. The homogenate was centrifuged at 2000 × g for 15 min and the PNS (post nuclear supernatant) fraction was layered on top of a Nycodenz gradient consisting of 3 ml each of ice-cold Nycodenz solutions of 28%, 19%, 7.3%, respectively, followed by centrifugation at 200,000 × g for 90 min at 4 °C in a Beckman SW41 rotor. The 19%/28% boundary (late endosome-rich fraction) was concentrated while changing the buffer HB150 using a VIVASPIN 2 (30,000 MWCO; Sartorius).

**Microinjection**. About 2 mM (proteo)liposomes and 10 μg/ml DAPI (4′,6-diamidino-2-phenylindole; Sigma) (injection marker) in HB150 were filled in Femtotips (Eppendorf); 1 × 10⁴ HeLa cells were plated on poly-L-lysine (Sigma-Aldrich)-coated 12 mm coverslips (Marienfeld GmbH) and then the coverslip was placed into a 35-mm petri dish (Becton Dickinson) filled with pre-warmed injection medium (F12 medium (Invitrogen), supplemented with 10% FCS, 10 mM HEPES

(pH7.5), and 100 units/ml each of penicillin and streptomycin). Microinjection was performed using a Femtojet (Eppendorf) and Injectman micromanipulator (Eppendorf) under a Leica DMIL inverted microscope for 5 min per coverslip. After microinjection, the cells were incubated at 37 °C in the cell culture medium and then fixed with 4% paraformaldehyde (Sigma-Aldrich) in PBS for 10 min followed by immunocytochemistry using antibodies specific for organelles as indicated. To label Tfn-positive endosomes, microinjection was performed in 5 μg/ml Alexa488-, Alexa568-, or Alexa633-Tfn-containing injection medium for 5 min. After microinjection, the cells were incubated for 5 min at 37 °C in the cell culture medium, then were fixed with 4% paraformaldehyde (Sigma-Aldrich) in PBS for 10 min. To take time-lapse images, injected cells were quickly set on an LSM 780 confocal microscope (Carl Zeiss).

**Image acquisition and processing**. Images were taken using an LSM 780 confocal microscope with a 63× water-immersion objective and Zen software.

The extent of co-localization between injected PLs and organelles was determined using a custom written Matlab algorithm (The Mathworks Inc.), kindly provided by Prof. Silvio Rizzoli (Göttingen, Germany). At least 100 injected vesicles from 3–15 independent experiments were analyzed for the co-localization with each organelle marker in an experiment of microinjection. Time-lapse images were taken at room tempreture using an LSM 780 confocal microscope (Carl Zeiss). Fluorescence intensity was calculated with Image J software and line scan assay was performed by Zen software. Calculation of the perinuclear index was performed as described in ref. [23]. In short, fluoresence intensity was measured for the whole cell except the nuclear area ($I_{total}$), the concentric half area from nucleus ($I_{perinuclear}$), and remaining area ($I_{peripheral}$). The perinuclear index was defined as Index = $I_{perinuclear}/I_{total} \times 100 - I_{peripheral}/I_{total} \times 100$.

**Immunoprecipitation and immunoisolation of membranes**. HeLa cells were washed in PBS and lysed in lysis buffer (50 mM Tris-HCl (pH 7.4), 150 mM NaCl, and 1% Triton X-100) containing Complete protease inhibitor cocktail (Roche) and centrifuged for 5 min at 4 °C to remove cell debris. The supernatants were incubated with the antibody at 4 °C overnight and incubated with Dynabeads Protein A or Dynabeads Protein G (Thermo Scientific) for 1 h. The beads were washed three times with lysis buffer and resuspended in gel-loading buffer. Samples were incubated for 10 min at 70 °C and analyzed by SDS-PAGE and immunoblotting. To detect the interaction between Rab GTPases and Vps13B, cells were cross-linked with DSP (Thermo Scientific) for 3 h and lysed with lysis buffer. For immunoprecipitation of GFP fused proteins, the supernatants were incubated with GFP-traps (ChromoTek) at 4 °C overnight. To immunoisolate vesicles carrying specific surface proteins, a PNS fraction was incubated with a pre-formed complex of the respective primary antibody and Protein G on Dynabeads at 4 °C overnight and washed three times with ice-cold PBS containing Complete protease inhibitor cocktail.

**Protein–lipid interaction assay**. For lipid–protein overlay assays, we used Echelon PIP strips (Echelon, P-6001; 100 pmoles/spot). The strips were blocked in 2% BSA in TBST (0.1% Triton X-100 in TBS) for 1 h at room temperature and incubated with cytosolic fractions from cells expressing GFP-Vps13B fusion proteins as indicated (final protein concentration was 10 μg/ml) for 3 h at 4 °C. They were washed three times with TBST, and protein bound to the lipids was detected by Western blotting with an anti-GFP antibody. For measuring binding of liposomes to Vps13B, magnetic Dynabeads containing Protein G were coated with anti-GFP antibodies and incubated with cytosolic fractions from HeLa cells expressing Vps13B-GFP or free GFP as control. After washing out unbound proteins, the beads were incubated with 0.3% Rhodamine-PE-containing liposomes with/without 5% PtdIns(3)P at 4 °C for 60 min. Precipitated liposomes were eluted with 1% Triton X-100 in HB150 buffer and fluorescence of Rhodamine in the buffer was measured with a fluorescence plate reader.

## Data availability
The data that support the findings of this study are available from the corresponding author upon reasonable request. The source data underlying Figs. 1c–e, 2a–c, 4a, d, and 7d are provided as a Source Data file.

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

## Acknowledgements

We are grateful to D. Czernik for assistance with mass spectrometry analysis, U. Ries for technical support, S. Rizzoli for kindly providing the Matlab algorithm for examining the co-localization (all Göttingen), and K. Nakayama (Kyoto) for providing mRFP-Rab14 constructs. We also thank A. Stein and H.D. Schmitt (Göttingen) for comments on the manuscript; S.K. was supported by the Uehara memorial foundation.

## Author contributions

S.K. carried out all experiments. S.K. and R.J. designed and discussed the experiments and wrote the manuscript.
