## [Peer Review File · Nature Communications]

Reviewers' comments:

Reviewer #1 (Remarks to the Author):

This paper presents a very interesting study that provides novel fundamental insights into the mechanisms underlying vesicle targeting specificity. The general view that emerged from decades of investigation was that the specificity of vesicle targeting and fusion involves at least two layers of protein-protein recognition, i.e. a first layer that involves binding of small Rab GTPases residing on the vesicle to effectors on the target membrane, most of which act as tethering factors, and a second layer involving in pairing of SNAREs between the two membranes, which provides some level of specificity in membrane fusion. In the past, there had been multiple reports of interactions between SNAREs and tethering factors, and a recent paper by the Jahn lab showed that injection of liposomes containing only SNAREs into cells leads to proper localization, indicating that, surprisingly, SNAREs alone are sufficient to dictate correct targeting. This new paper by the Jahn lab now reveals that SNARE-tethering factor interactions contribute dramatically to vesicle targeting specificity. Moreover, specificity involves combinatorial effects arising from distinct interactions between SNAREs and tethering factors. In particular, they show that vesicles containing syntaxin-13 and syntaxin-6 are targeted to a different compartment than vesicles containing only syntaxin-6, and these differences depend on two tethering factors, Vps51 and Vps13B. The paper also provides new insights into the function of Vps13B, a protein that had not been sufficiently studied.

I believe that this paper will be highly influential in the field of membrane traffic and I strongly recommend publication in Nature Communications. I only have two minor concerns:

1. Although I find the results very convincing, this work focuses on a subset of SNAREs and tethering factors. Hence, the generality of the new concept on specificity will need to be confirmed with further studies and I encourage the authors to tone down a little bit the main conclusions.
2. The data shown in the paper in general exhibit the high quality that is common in the papers from the Jahn lab, but some of the gels are of rather poor quality (e.g. Figs. 6a, 6d and S3a), which may make the specific conclusions from those experiments less convincing to the reader.

Reviewer #2 (Remarks to the Author):

SNARE proteins play a key role in cell organization by pairing selectively to cognate SNAREs via their SNARE domains. This study addresses the question of whether SNARE proteins also contain information which may impact targeting of vesicles that harbor them in their membrane, independently of their property to pair with other SNAREs. To do so, they build on the experimental model that they have previously developed, the microinjection in cells of vesicles in which they have reconstituted specific SNAREs and the analysis of the targeting of these vesicles in host cells. By focusing on stx6 and stx13 the authors find that this is the case. They additionally identify VPS51 and the very large protein VPS13B as cellular factors required for the targeting of these SNAREs to endosomes using a variety of complementary approaches. They subsequently focus on Vps13B. They show that Vps13B knockdown results in a mistargeting of internalized Tfn and thus argue that Vps13B a tethering factor operating in the Tfn recycling pathway. They also demonstrate that VPS13B bind to stx6/stx13, to a subset of Rabs and to PI3P. Based on these findings, they draw the conclusion that VPS13B may be the first representative of a novel and class of tethering proteins.

The identification of mechanisms that help direct vesicles containing specific SNAREs to their destination is a result of broad interest in cell biology. I find this part of the study solid and convincing. The work on VPS13B is more preliminary. Suggestions for improvement include the

following:

1. More information should be provided on the identification of Vps13B by mass spectrometry. Can the authors present a table of proteins identified showing how Vps13B ranks in this interactome? Others have reported that Vps13B is not found in appreciable quantities in the cytosolic fractions. Given that cytosolic proteins were used for identification, reporting this information would be helpful.

2. Binding of VPS13B to PI3P in the PIPstrip experiment can be very indirect. If the authors want to state that Vps13B is a PI3P interactor, they should provide additional data for this interaction, or acknowledge the limitations of this experiment.

3. Are the interactions of Vps13B with Rabs GTP-dependent? One could draw the message that Vps13B is an effector of a subset of Rabs, but this would require this interaction to be GTP dependent.

4. The discussion of Vps13 should be improved by better incorporating recent findings suggesting that Vps13B may be a lipid transport protein.

Minor: Is the calibration bar of fig. 4e correct? Mitochondria have roughly the width of a bacterium and that bar does not seem to be consistent.

Point-by-point response:

Reviewer #1,

1. Although I find the results very convincing, this work focuses on a subset of SNAREs and tethering factors. Hence, the generality of the new concept on specificity will need to be confirmed with further studies and I encourage the authors to tone down a little bit the main conclusions.

Answer

As requested by the reviewer, we have rephrased some of our conclusions in order to acknowledge that further confirmation of the concept is required (see Page 5 line 121, and Page 15 line 438, 449-455).

2. The data shown in the paper in general exhibit the high quality that is common in the papers from the Jahn lab, but some of the gels are of rather poor quality (e.g. Figs. 6a, 6d and S3a), which may make the specific conclusions from those experiments less convincing to the reader.

Answer

As requested by the reviewer, we have repeated some of the experiments and replaced the panels of Fig. 6a and 6d (now 6e in the revised version) and S3a with new data.

Reviewer #2,

1. More information should be provided on the identification of Vps13B by mass spectrometry. Can the authors present a table of proteins identified showing how Vps13B ranks in this interactome? Others have reported that Vps13B is not found in appreciable quantities in the cytosolic fractions. Given that cytosolic proteins were used for identification, reporting this information would be helpful.

Answer

As requested by the reviewer, we have included the result of the mass spectrometry analysis in Supplementary table 1.

Moreover, for the information of the reviewer, we include an immunoblot showing the distribution of Vps13B after fractionation by high-speed centrifugation of HeLa cell lysates (equal amounts of protein were loaded). Note that Vps13B is present in both fractions. As control, we used Na⁺/K⁺-ATPase for membrane marker and tubulin as control for a soluble protein (extraction was carried out under conditions leading to depolymerization of tubulin). This experiment confirms that a sizeable pool of Vps13B is present in the cytosol.

2. Binding of VPS13B to PI3P in the PIPstrip experiment can be very indirect. If the authors

want to state that Vps13B is a PI3P interactor, they should provide additional data for this interaction, or acknowledge the limitations of this experiment.

Answer

To address this issue, we have carried out a complementary experiment to show the interaction between Vps13B and PtdIns(3)P (new Fig. 6f). We first immobilized Vps13B-GFP on anti-GFP antibody coated magnetic beads by incubating the beads with cell extracts transfected with Vps13B-GFP, followed by extensive washing to remove adherent proteins. The beads were then incubated with liposomes containing PtdIns(3)P and labeled with Rhodamine-PE, and bound liposomes were quantified by measuring Rhodamine fluorescence of Rhodamine. These data support our conclusion. Note that we cannot completely exclude that additional proteins remain bound to Vps13B, and this is now explicitly acknowledged in the text.

3. Are the interactions of Vps13B with Rabs GTP-dependent? One could draw the message that Vps13B is an effector of a subset of Rabs, but this would require this interaction to be GTP dependent.

Answer

To address this issue, we have transfected HeLa cells with mRFP-tagged versions of GTP-preferring (QL) or GDP/nonbinding (SN) form of Rab14, followed by IP of the Rab variants using an anti-GFP antibody, followed by probing the immunoprecipitates for Vps13B (Fig. 6c). The result suggests that the interaction between Vps13B and Rab14 is independent of the GTP/GDP status of Rab14.

4. The discussion of Vps13 should be improved by better incorporating recent findings suggesting that Vps13B may be a lipid transport protein.

Answer

We have revised the discussion for better covering the function of Vps13 as a lipid transporter (from page 17 line 503) and its dual function in lipid transport and tethering/fusion (page 17 line 516 onwards).

Minor: Is the calibration bar of fig. 4e correct? Mitochondria have roughly the width of a bacterium and that bar does not seem to be consistent.

Answer

The scale bar is correct. As the reviewer suggested, the width of single mitochondrion is normally less than 1 μm . However, the width of mitochondria in Fig. 4e is approximately 1-2 μm . We think the mitochondria is not a single, rather some mitochondrion aggregates closely together. We can see that GFP-MAO labeled mitochondria are aggregated in lower

magnification of images (supplementary fig. S3e). Such aggregation was observed in some experiments, but we could not know why aggregation happens. Importantly, the aggregation of mitochondrion did not affect the association of liposomes to mitochondria.

REVIEWERS' COMMENTS:

Reviewer #1 (Remarks to the Author):

The authors have addressed the reviewer concerns satisfactorily and I recommend publication of this paper in its current form.

Reviewer #2 (Remarks to the Author):

The authors have satisfied my questions and the manuscript can now be accepted for publication.